# “I Think That’s the Most Beneficial Change That WIC Has Made in a Really Long Time”: Perceptions and Awareness of an Increase in the WIC Cash Value Benefit

**DOI:** 10.3390/ijerph19148671

**Published:** 2022-07-16

**Authors:** Emily W. Duffy, Daniele A. Vest, Cassandra R. Davis, Marissa G. Hall, Molly De Marco, Shu Wen Ng, Lindsey Smith Taillie

**Affiliations:** 1Carolina Population Center, Department of Nutrition, University of North Carolina at Chapel Hill, Chapel Hill, NC 27599, USA; shuwen@unc.edu (S.W.N.); taillie@unc.edu (L.S.T.); 2Department of International Health, Johns Hopkins Bloomberg School of Public Health, Baltimore, MD 21205, USA; dvest3@jhu.edu; 3Carolina Population Center, Department of Public Policy, University of North Carolina at Chapel Hill, Chapel Hill, NC 27599, USA; cnrichar@email.unc.edu; 4Carolina Population Center, Department of Health Behavior, University of North Carolina at Chapel Hill, Chapel Hill, NC 27599, USA; mghall@unc.edu; 5Department of Nutrition, University of North Carolina at Chapel Hill, Chapel Hill, NC 27599, USA; molly_demarco@unc.edu

**Keywords:** fruit, vegetable, childhood, COVID-19, cash value benefit

## Abstract

During the COVID-19 pandemic, the Special Supplemental Nutrition Program for Women, Infants, and Children (WIC) Cash Value Benefit (CVB) for fruits and vegetables increased by roughly USD 25/month/person. We sought to understand WIC participant perceptions of this change and barriers and facilitators to using the CVB. We conducted 10 virtual focus groups (5 rural, 5 urban/suburban) with WIC participants (*n* = 55) in North Carolina in March 2022. Focus groups were recorded and transcribed. We open-coded the content and used thematic analysis to uncover consistencies within and between sampled groups. Participants expressed favorable perceptions of the CVB increase and stated the pre-pandemic CVB amount was insufficient. Barriers to using the increased CVB were identifying WIC-approved fruits and vegetables in stores and insufficient supply of fruits and vegetables. Barriers were more pronounced in rural groups. Facilitators of CVB use were existing household preferences for fruits and vegetables and the variety of products that can be purchased with CVB relative to other components of the WIC food package. Participants felt the CVB increase allowed their families to eat a wider variety of fruits and vegetables. The CVB increase may improve fruit and vegetable intake, particularly if made permanent, but barriers to CVB and WIC benefit use may limit the potential impact.

## 1. Introduction

Consuming a sufficient amount and variety of fruits and vegetables in early childhood is critical to forming lifelong health-promoting dietary habits [1,2]. A nutritionally adequate diet in early childhood is key for optimal physical and cognitive growth and development [3,4]. Fruits and vegetables are key sources of nutrients commonly under-consumed by young children in the US, and they reduce the lifetime risk of chronic health conditions [5,6,7,8,9]. Consumption of fruits and vegetables, especially nutrient-dense varieties, is often lower among children living in rural households and households with low incomes and children from historically marginalized racial or ethnic groups [10,11,12,13,14,15,16]. Across the US, structural factors such as high cost and disparate physical access to fruits and vegetables as well as divestment in communities make it more challenging for children living in rural areas, in households with low incomes, and from historically marginalized racial/ethnic groups to meet fruit and vegetable intake recommendations [10,11,12,13,14,15,16,17,18,19,20,21,22,23,24,25]. In the rural Southeastern US, these geographic, income, and race/ethnicity groups often intersect and overlap, contributing to a potentially greater risk of inadequate fruit and vegetable intake [17].

Historically marginalized communities are disproportionately impacted by public health emergencies such as pandemics and natural disasters [26,27,28,29,30]. The COVID-19 pandemic has followed a similar pattern: families living in rural areas with low incomes and from historically marginalized racial/ethnic groups in the US have been more likely to experience job loss and nutrition insecurity because of the pandemic [25,31,32,33,34,35,36,37]. These downstream effects of the pandemic have the potential to exacerbate disparities in fruit and vegetable consumption by income, race/ethnicity, and rurality. The pandemic has also created food supply chain issues, including widespread food shortages and rising food costs due to inflation [38,39]. These issues may have disproportionately impacted people living in rural areas: even prior to the pandemic, many factors such as food cost and access to emergency food programs were more notable barriers to achieving a healthy diet in rural areas compared to urban areas [17,40]. Thus, it is important to understand differences in the effects of the pandemic on diet-related behaviors and disparities by rurality.

To counteract some of the negative effects of the pandemic on nutrition security, the United States Department of Agriculture (USDA) implemented a series of modifications and augmentations to its existing federal nutrition assistance programs [41]. The Cash Value Benefit (CVB) is a component of the food package for the Special Supplemental Nutrition Program for Women, Infants, and Children (WIC) that can be used for fresh, frozen, or canned fruits and vegetables without added sugar, salt, or fat. Prior to the pandemic, the CVB was USD 9–11/month/person, an amount that many WIC participants and nutrition experts deemed insufficient [42,43,44,45]. In June 2021, the USDA temporarily increased the CVB to USD 35/month/person, initially for four months. Ultimately, this increase was extended until September 2022, but at slightly different amounts (Figure 1).

Preliminary quantitative research on the CVB increase generally suggests that it has been positively received by WIC participants and may be associated with an increased intake of fruits and vegetables [46]. Additional qualitative studies can complement this existing research by exploring WIC participants’ lived experiences with the CVB increase. Moreover, to understand the potential public health benefits of the CVB increase and to inform future changes to the WIC food package, it is essential to understand WIC participants’ awareness of the change, barriers, and facilitators to using the higher CVB amount, and perceived changes in dietary behaviors. However, studies have not yet explored these questions or examined differences in experiences based on rurality. This information is critical for developing evidence-based public health emergency response policies as well as informing discussions about extending the higher CVB amount beyond September 2022.

The primary objectives of our study were to qualitatively examine (1) perceptions and awareness of the CVB increase, (2) barriers and facilitators to using the increased CVB, and (3) perceived effects of the CVB increase on household fruit and vegetable consumption. We also aimed to understand whether experiences and perceptions of the CVB increase differed by rurality, given disparities in food costs, food environments, and downstream effects of the pandemic that may affect CVB use [17,23,24,31]. Finally, we examined facilitators and barriers to WIC benefit use beyond just the CVB component since any barrier to general WIC benefit use could, in turn, influence CVB use.

## 2. Materials and Methods

*Sample:* In February and March of 2022, we recruited 55 WIC participants living in North Carolina for virtual focus groups. To be eligible for the focus groups, participants had to be 18 years or older, enrolled in WIC any time after May 2021, take part in household grocery shopping, speak English, identify as a woman, live in North Carolina, not be an employee of WIC, and have access to Wi-Fi or a cell phone signal strong enough to participate in the Zoom call. We decided to not enroll men in our study given the small number of men who are the primary caregiver for children participating in WIC in NC and because we wanted to create focus groups with individuals that share identities to facilitate sharing [47]. We stratified the focus groups by rural and urban/suburban residents. We categorized North Carolina’s 100 counties using the North Carolina Rural Center’s definitions (6 urban, 16 suburban, 78 rural) [48].

*Recruitment:* We partnered with the North Carolina Department of Health and Human Services (NC DHHS) and local WIC agencies across North Carolina to recruit participants. These agencies shared information about our study on their social media pages and through mailed flyers and flyers in clinics. We also shared information about our study with the statewide network of Supplemental Nutrition Assistance Program Education (SNAP Ed) agencies and through the local organizations that were part of our stakeholder advisory board. Interested participants completed an online screening questionnaire which was programmed into Qualtrics. We also applied additional criteria to screen out potentially fraudulent participants (e.g., individuals who did not live in the U.S. or have a child but were misrepresenting this information). These criteria included confirming that IP addresses were in North Carolina, matching responses to duplicate questions about age, using Qualtrics’s bot detection item, preventing duplicate submissions, and screening out responses based on Qualtrics’s fraud detection scores [49]. Additionally, we conducted brief screening Zoom calls with each participant who was deemed eligible based on both screener questions and Qualtrics meta-data to confirm eligibility and troubleshoot any issues with Zoom connectivity prior to the focus group discussions. Previous studies used similar multistep approaches to improve screening for qualitative research [50,51]. Written informed consent was collected electronically from all participants. This study was reviewed and deemed exempt from further review by the University of North Carolina Institutional Review Board (IRB #21-2873).

*Procedures:* We collected demographic information from participants in the screening questionnaire. We conducted 10 virtual focus groups using Zoom in March of 2022. Focus groups were facilitated by one of two graduate students (EWD, DAV) trained in focus group facilitation techniques. Each focus group had four to eight participants, and, when possible, groups were composed of participants of similar race and ethnicity to facilitate sharing and create comfort while discussing potentially sensitive topics [47]. Between 50–100% of participants that signed up for a focus group discussion slot attended on the day of the discussion. To measure race/ethnicity, we used two items using self-classification [52] from the 2020 United States Census Bureau [53]. We used participants’ responses to these items to create a race/ethnicity variable combining the self-classified race with Hispanic, Latina, or Spanish origin (Table 1). We conducted five focus groups each among rural and urban/suburban participants, and we reached a point of saturation in each subgroup [47]. We assessed saturation by determining that we heard the same themes repeatedly and no new codes were being developed [54]. Each focus group lasted approximately one hour, and participants received a USD 40 gift card for their time.

We used a semi-structured focus group guide for all discussions. This guide was developed in consultation with our stakeholder advisory board and NC DHHS. We also used prior research related to WIC grocery shopping and CVB use experiences to ensure that our questions aligned with relevant content. In North Carolina, the CVB amounts increased and decreased at multiple time points between June 2021 and our study period due to the timing of congressional decisions and a change from USD 35/month/person to amounts recommended by the National Academies of Science, Engineering, and Medicine (NASEM) (USD 24/month for children 1–5 years, USD 43/month for pregnant and postpartum participants, and USD 47/month for breastfeeding participants) (Figure 1), so we were interested in participants’ experiences with these changes over time. This change to the NASEM amounts was an increase for some families and a decrease for others, depending on household composition. The guide assessed: perceptions and awareness of the CVB increase, barriers and facilitators to using CVB at the higher amounts, perceived changes in household dietary behaviors, general barriers and facilitators to using WIC benefits and how that may have changed during the pandemic, and perceptions of the WIC food package (Appendix A).

All focus groups were recorded and transcribed using Otter artificial intelligence transcription software [55]. If participants shared ideas in the Zoom chat, we incorporated their chat comments into the transcript. Either EWD or DAV double-checked the accuracy of transcripts and provided edits when needed using the recordings. Transcripts were not reviewed by participants, but a summary of key study findings was shared with participants. This process did not result in more data or change the interpretation of results.

*Data analysis:* Focus group transcripts were analyzed using thematic analysis based on a phenomenological approach, which is used to study how people make meaning of their lived experiences [56]. We deemed this approach suitable for data analysis, given our interest in assessing participants’ experiences with the pandemic and the CVB increase. An initial codebook was developed a priori based on relevant research from relevant topics. After reading through (without coding) a random sample of three of the transcripts, we updated the codebook and refined emergent codes. All authors provided input on the codebook. Then, three transcripts were double-coded by EWD and DAV and the codebook was updated and refined after each transcript was reviewed (Appendix A). EWD coded the remaining seven transcripts using the revised codebook. Based on these analyses, codes were aggregated into themes and memos were developed summarizing findings from each key theme. The coding density of each theme was examined among the rural and urban subgroups to identify similarities and differences. All coding and analyses were conducted using NVivo [57]. We used the COREQ checklist to ensure comprehensive and transparent reporting of our methods [58].

*Positionality and Reflexivity*: It is important to acknowledge our research team’s positionality. Our team has lived experiences and social identities that are similar to and different from our study participants. These identities can influence the way that we developed our research questions, wrote our focus group guide, facilitated focus group discussions, and analyzed and presented our results [59,60]. For example, the lead author [EWD] is a white woman that does not have lived experiences with federal food assistance programs, is not a parent or caregiver, and her primary research interest is in nutrition policies that affect early childhood nutrition. Although the study team used numerous measures to account for differences in our team and lived experience of our study participants, it is possible that these identities and interests influenced the types of questions that we asked (e.g., we may have missed important questions about using WIC due to lack of experience) or the way we presented results (e.g., selection of quotes). Throughout the data collection and analysis process, we examined and questioned our pre-existing beliefs with the goal of identifying ways in which these beliefs could have influenced study results [59,60]. Additionally, in an effort to account for differences in lived experiences and identities, we developed a stakeholder advisory board with WIC staff and community organizations representing individuals with similar lived experiences to our participants and sought this board’s input at each step of the research process.

## 3. Results

### 3.1. Participant Demographics

We had 55 participants in our 10 virtual focus groups, 29 in the urban focus groups and 26 in the rural focus groups (Table 1). The average age of mothers or caregivers was 30.4 years. Forty-two percent of participants reported an annual household income of USD 24,999 or less, and 50% reported a household income between USD 25,000 and USD 49,999. Among all participants, 42% of the sample was Non-Hispanic/Non-Latina Black or African American, 24% were Hispanic or Latina, and 24% were Non-Hispanic/Non-Latina White. About half (51%) of participants reported currently participating in the Supplemental Nutrition Assistance Program (SNAP). About one-quarter (22%) of participants had a high school education or less, and roughly half (53%) had some college education or an associate degree. On average, participants from the urban groups were older, had higher incomes, had higher levels of education, and were more likely to participate in SNAP (Table 1).

### 3.2. Themes from Focus Groups

Below, we present findings based on pertinence to the CVB policy change, how topics were organized in our focus group guide, as well as our primary research questions. We structured our themes around the key topics of the focus group guide because these questions were designed to address specific gaps related to a policy change [61]. The main themes that emerged from the focus group discussions were perceptions of the CVB amounts before and after the pandemic, awareness and lack of awareness of CVB increase, barriers and facilitators to using CVB, barriers and facilitators to using WIC benefits in general, and desired changes to CVB and the WIC food package. These themes and relevant subthemes are described below and summarized in Appendix A.

#### 3.2.1. Cash Value Benefit Increase

##### Perceptions of Pre-COVID CVB Amount

Overall, participants expressed that the CVB amount before June 2021 (USD 9–11/month/person) was insufficient. They described how this amount usually lasted for only one week and limited the varieties of fruits and vegetables they could buy. Many participants turned to more shelf-stable, low-cost fruit and vegetable varieties such as bananas, a bag of oranges, or canned items to make the amount stretch. Additionally, some participants felt this amount was insulting because it was so low and inconsistent with the nutrition advice provided by the WIC, which encourages parents and their children to consume a large number of fruits and vegetables. For example, one participant stated:
*I remember asking the nutritionist, like, “Why do you only give this small amount?” And she started trying to tell me about how “Oh, well the purpose of the WIC program is to be able to, you know, combine the different foods. So like, you can use a little bit of the fruits for like a smoothie and this and that.” And I just remember feeling like not, not like I had any agency in deciding like how I wanted my diet to be…*

##### Awareness and Perceptions of the CVB Increase

Some participants were notified by their local WIC agency about the initial CVB increase in June 2021, the subsequent decrease in North Carolina in October 2021, and the changes to the NASEM amounts in November 2021. Rural participants were more likely to report receiving notification about the CVB changes as compared to their urban peers. However, many participants were not notified by their local WIC agency about these changes and found out by checking their WIC benefit balance on their BNFT app, during checkout, or from their grocery store receipts. Participants also mentioned how the changes in the CVB amount over time made it difficult to plan for meal preparation based on their available benefits as they normally would. This lack of awareness also created some challenges and uncertainty among participants about the accuracy of their WIC balance and the duration of the increased CVB. For example, several participants did not know about the one-month decrease until they went to check out at the grocery store and then described having to put things back, use SNAP, or pay with their own funds to cover the difference. One respondent shared:
*I wish I would got a text or a call from one of the representatives and be like “Hey, this month, we’re gonna be cutting back on some of your, your money for fruit. We just wanna let you know.” Instead of me going to store and me looking crazy ‘cause I’m finna buy all this fruit and I can’t ‘cause I ain’t got enough money.*

Despite these implementation challenges, participants had favorable perceptions of the CVB increase. Participants expressed some dissatisfaction with the initial increase to USD 35/month/person and the later decrease to USD 24/month/person for children. For example, one participant stated:
*But now they can kind of dwindled it back down or whatever to only like 20 something dollars and it’s just like “But why though?*

Participants also agreed that the CVB was one of the most valuable components of the WIC food package. They noted that they often spend use their CVB first before other WIC food package components (e.g., beans, cereal) each month and that the CVB was the component that needed to be increased the most during the pandemic. One participant stated:
*…the fruits and vegetables I think is like the most important thing. And I think that is more important than eggs, more important than milk, more important than cereal. They all have their benefits. But I think the vegetables, especially if you’re going to start the kids off when they’re young, you have to give them the vegetables when they’re little or they’re not going to want them.*

Some mentioned that the CVB increase influenced their decision to remain enrolled in the WIC program. Participants shared that fruits and vegetables are a pivotal part of being able to provide healthy meals for their family, that their families enjoy eating and prefer fruits and vegetables, and that this benefit increase allowed their families to achieve dietary patterns more closely aligned with their family’s preferences and WIC recommendations. These perceptions were similar across the rural and urban groups.

##### Barriers and Facilitators to Using CVB

Most participants reported that they used the full amount of their CVB each month, and many mentioned they go through the current (NASEM) amount on their first trip to the grocery store after their benefits are renewed. These perceptions were similar among rural and urban participants. Participants felt they needed more than the current CVB amount to meet their family’s needs, especially since the average cost of fruits and vegetables has increased with inflation and the CVB is the only dollar-value-based component of the WIC food package. One participant stated:
*…everything costs so much more, your $9 that would have gotten you, would have gotten you a lot more last summer than it’s going to get you this summer…they also need to think about the reality of inflation and so that like what we can actually get is actually smaller…*

Participants mentioned that not being able to scan certain fruit and vegetable products in the BNFT app presented a challenge, particularly when produce was not clearly labeled as WIC approved. Participants also described issues at checkout when fruits and vegetables they thought would be covered by WIC, such as frozen fruit, were not, and they had to pay out of pocket for these products. Barriers to using the CVB were more pronounced among rural participants compared to urban participants. Rural participants often highlighted a lack of adequate supply of fruits and vegetables in grocery stores. The more general WIC use barriers discussed below, such as the time and mental burden of using WIC benefits and lack of desired technologies, such as online shopping and self-checkout, are also important barriers to CVB use.

Despite these barriers, most participants felt it was easy to use the full CVB amount each month because of the variety of products (e.g., fresh, canned, or frozen fruits and vegetables) that could be purchased with CVB. Participants also said that it was easy to spend the full amount because their families preferred to eat fruits and vegetables, and they are part of their day-to-day meals. Finally, participants mentioned certain grocery stores or places such as farmers markets with fruit and vegetable incentive programs that had appealing and fresh produce that made it easy for them to use their CVB each month. Participants in rural and urban areas had similar perceptions of what factors facilitate their use of the CVB.

##### Perceived Changes in Household Food Behaviors

Participants believed the CVB increase allowed their families to eat healthier. They also stated the CVB increase allowed them and their children to eat a wider variety of fruits and vegetables and allowed their children to try new fruits and vegetables. One participant said:
*And we’ve discovered that he loves asparagus and broccoli. So, we could like do that for lunch or like a little midday snack. I give him some grapes, and like broccoli, or strawberries, and asparagus, just for a healthier snack or lunch, instead of going to like freezer meals and potato chips and stuff like that.*

This theme of increased variety was common among rural and urban participants but more pronounced among urban participants. Participants also said the CVB increase allowed them to introduce new fruit and vegetable varieties without the fear of wasting food that they had when the CVB was lower. Participants also felt the CVB increase led to a change in their dynamic with their children while grocery shopping. For example, children would ask for new varieties of fruits, and participants were able to buy these products for their children for the first time.

#### 3.2.2. General WIC Benefit Use

##### Facilitators and Barriers to Using WIC Benefits in General

Clear and accurate labeling at the point of selection of which products were WIC-approved was a key determinant of which stores participants preferred to use their WIC benefits in and a facilitator to using WIC benefits. Many participants also mentioned the transition from paper vouchers to the electronic benefit transfer (EBT) system has made using WIC benefits much easier. Some participants, and urban participants especially, also stated the WIC BNFT smartphone app made it easier to identify WIC-approved products. Participants stated that, during the pandemic, in particular, the flexibilities implemented by WIC in the food package, such as substitutions of products within a category and remote/phone appointments, supported their use of WIC benefits, and they wanted these flexibilities to remain in place beyond the pandemic [62].

Despite some retailers having clear and accurate labeling, participants mentioned significant barriers to identifying WIC-approved products in most retailers due to non-existent or inaccurate labeling, which deterred them from using WIC benefits at these outlets, sometimes despite more competitive pricing. Similarly, participants mentioned issues at checkout due to incorrectly labeled WIC-approved items they thought were approved. Participants also discussed the time and mental burden of using WIC benefits compared to other payment types, such as challenges remembering which products were WIC-approved, having to go to multiple stores to find WIC-approved items due to shortages, and remembering to use all their WIC benefits before they expire each month. Some participants also mentioned the stigma associated with using WIC and experiencing issues at checkout and coping mechanisms to avoid this stigma, such as shopping at less popular times of the day. Delays in receiving benefits due to limited staffing, unpleasant interactions with WIC staff, and lack of culturally relevant items in the food package also presented barriers to WIC use.

One of the most notable barriers to using WIC was the desire for new technologies, such as the ability to use WIC at self-checkout or for online grocery shopping. This was particularly true during the pandemic. Participants described the inconvenience of not being able to use WIC for online shopping. They described the fear they often had going into grocery stores to use their WIC benefits because they did not want to risk exposure to COVID-19 for themselves or their children. Shortages, particularly milk, lactose, free milk, and infant formula, presented challenges to using WIC benefits during the pandemic. These shortages were particularly common among participants living in rural areas. Additionally, participants noted higher food costs presented challenges for their families and sometimes contributed to food insecurity, particularly in rural participants. Each of these barriers to using WIC benefits, in general, can also be considered barriers to using the CVB component of WIC benefits.

#### 3.2.3. Desired Changes to CVB and the Food Package

When asked about suggested changes to the CVB, participants wanted to continue to receive this benefit for their 6–12-month-old children once complementary foods were introduced so that they could make their own pureed baby foods instead of receiving the jarred baby foods. They also stated they needed more than the current NASEM recommended amounts for fruits and vegetables to provide adequate fruits and vegetables for themselves and their children. Participants were also interested in the idea of being able to substitute components of the WIC food package across and within categories or personalize the food package to better suit their family’s and children’s preferences. One participant stated,
*…if I could say, you know, you can keep this bread and give it to someone who would actually use this bread and someone who will actually use this cereal, go ahead and just give me $5 more for fruits and vegetables, and that would be fine. Like, I just think if it’s like tailored to the child like that…*

Participants also wanted their WIC benefits to roll over for at least one month, similar to how SNAP benefits are administered. Many participants mentioned the current means of administering WIC benefits one month at a time created anxiety about forgetting to use benefits before they expired. Additionally, some participants stated that rolling over benefits would allow them to better meet their young children’s constantly evolving food preferences. Urban participants tended to suggest more changes to the CVB amounts and WIC food package. Rural participants had fewer suggested changes, and some made statements such as “*I’m in no place to argue with them* [WIC administrators]” when asked about desired changes to the WIC food package.

## 4. Discussion

Through this qualitative study, we found that, among North Carolina WIC participants, the CVB increase was positively perceived, the pre-pandemic CVB amount was insufficient to meet WIC participants’ needs, and participants believed the CVB increase improved their households’ total fruit and vegetable consumption and increased the variety of fruits and vegetables consumed. However, despite these positive changes, we observed barriers to CVB and WIC benefit use, including lack of physical access and challenges identifying WIC-approved products. There were a few key areas in which rural and urban participants differed, as described further below, but overall experiences with the CVB increase were relatively similar between the two subgroups.

Our findings that participants perceived improvements in fruit and vegetable consumption following the CVB increase are consistent with a recent report [63] which also noted that CVB increases allowed WIC families to consume more fruits and vegetables and a wider variety of fruits and vegetables. Larger, quantitative studies with food purchasing or WIC redemption data will be needed, but our findings suggest the CVB increase may have improved fruit and vegetable intake in households with low incomes, from historically marginalized racial/ethnic groups, and in rural households, suggesting the promise of the CVB increase for mitigating disparities in fruit and vegetable intake in these populations. Additionally, repeated exposure to a variety of fruit and vegetable flavors and textures in early childhood is critical to developing a preference for these food groups [1]. However, the cost of this repeated exposure and the associated food waste is a barrier for families with low incomes to introduce young children to new foods they may not readily accept [64,65]. There was a consensus among participants in our study that this CVB increase allowed them and their children to try fruits and vegetables they had never been able to purchase before because they were cost prohibitive or because they feared wasting food. Beyond simply measuring total fruit and vegetable consumption, future studies should also examine the variety of fruits and vegetables consumed or purchased before and after this policy change.

Participants highlighted several barriers to using the CVB specifically and discussed a variety of more general barriers to using WIC benefits which, in turn, present barriers to using the CVB component of the food package. Participants in our study described barriers such as inaccurate labeling and issues at checkout with fruits and vegetables being deemed ineligible that they thought were eligible, similar to what prior research has consistently documented [66,67,68]. This barrier is not unique to the CVB and appears to be more of an issue with redeeming WIC benefits in general. Similar to prior studies documenting WIC shopper experiences [23,66,68,69,70], our study highlighted several general WIC use barriers, such as issues with stigma and lack of desired technologies that participants felt affected their WIC and CVB redemption. In our study, WIC participants also described various forms of what Elliot et al. described as disenfranchisement (i.e., structures that keep people from seeking public resources [25]), such as experiencing delays in receiving their WIC benefits due to staff shortages in rural areas, being afraid or hesitant to access benefits due to the risk of contracting COVID-19 or unpleasant interactions with WIC staff, and lacking access to fruits and vegetables or other foods in their communities. Additionally, we found that changes in the CVB amount over the period of June to December 2021, including a one-month temporary *decrease* in benefits, created a notable amount of confusion and uncertainty about redeeming CVB among North Carolina participants. These challenges are similar to the learning costs [71,72] associated with public assistance programs that present major barriers to use, and these barriers should be considered by policymakers when designing future emergency food response programs. Overall, there are still a variety of barriers to using the CVB and WIC benefits more generally that urgently need to be addressed for WIC to have the greatest possible impact on reducing diet-related disease and fruit and vegetable consumption disparities by income, race/ethnicity, and rurality.

To our knowledge, this is the first study to examine differences in experiences with the CVB increase by WIC participant rurality. Contrary to our expectations, despite some reported WIC staff shortages in rural areas, rural participants more commonly reported being told by their local WIC agency about some of the CVB changes compared to urban participants. We found that rural participants reported CVB and WIC use barriers such as unclear labeling, issues with the BNFT app, and a desire for self-checkout or online shopping. Others described the potential promise of online grocery shopping to alleviate food access issues in rural areas [73,74], but there continues to be low availability of online grocery options in rural areas compared to urban areas [75,76]. WIC is slated to be approved for online grocery shopping in the near future [77], so particular attention should be paid to uptake in rural communities. Consistent with other studies in NC describing challenges with healthy food access in rural communities [23,24], rural participants, in particular, noted that food supply issues such as a lack of fresh, culturally appropriate, and appealing fruits and vegetables presented a barrier to using their CVB and this was exacerbated by shortages experienced because of the pandemic. Some studies suggest that rural communities may have been disproportionately impacted by many aspects of the pandemic [31,32], as is true with most public health emergencies. Future studies should continue to examine the disparate effects of COVID-response programs in rural and urban communities as this could inform whether differential supports are needed long-term and in future emergencies. However, our results and reported differences by rurality should be interpreted with caution as this was a small, qualitative study in one state, and larger, more representative studies will be needed.

The strengths of this study include partnering with state and local-level stakeholders throughout the research project and timing the focus groups shortly after a policy change to capture responses when they were fresh in participants’ minds. Additionally, we successfully recruited a sample that was racially and ethnically diverse as well as reached saturation of themes among rural and urban/suburban participants; therefore, the perspectives described represent a wide variety of experiences. That being said, our sample size is relatively small and only represents the perspectives of North Carolinians are reflected in this study, so future studies using national samples and food consumption or purchasing data will be needed to more fully understand the effects of this policy change on WIC participants. Additionally, we were not able to adequately represent Hispanic/Latina WIC participants as we only were able to offer focus groups in English due to resource constraints. Given our recruitment strategies and the use of virtual focus groups, our sample likely reflects WIC participants that are more technologically savvy, have better cell phone service or Wi-Fi access, and are less hesitant about interacting with institutions such as universities. Finally, the use of one coder for the majority of the transcripts can be considered a limitation as this coder’s positionality may have influenced the interpretation of results.

## 5. Conclusions

Participants in our qualitative study had generally favorable perceptions of the pandemic-related CVB increase. Participants perceived that it improved their household’s total fruit and vegetable consumption and increased the variety of fruits and vegetables consumed by caregivers and their children but reported barriers to CVB and WIC benefit use must be addressed. The effects of the pandemic on nutrition security among households with low incomes will likely persist for years [78], so public health and social support policies such as this CVB increase may be a promising strategy for increasing access to fruit and vegetables and mitigating the negative effects of the pandemic on diet-related disparities.

## Figures and Tables

**Figure 1 ijerph-19-08671-f001:**
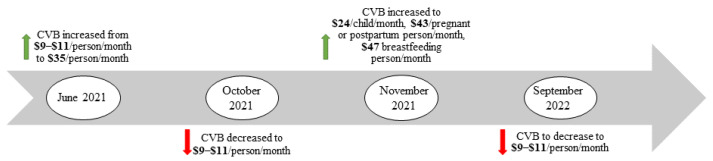
Timeline of key CVB changes between 2021 and 2022 in North Carolina.

**Table 1 ijerph-19-08671-t001:** Sample Demographic Characteristics (*n* = 55).

	Rural (%) (*n* = 26)	Urban (%) (*n* = 29)	Total (%) (*n* = 55)
**Average age**	29.2	31.6	30.4
**Race/Ethnicity ***			
Hispanic or Latina	4 (15)	4 (14)	8 (15)
Black or African American	7 (27)	16 (55)	23 (42)
White	9 (35)	4 (14)	13 (24)
Asian	0 (0)	1 (3)	1 (2)
Middle Eastern or North African	0 (0)	1 (3)	1 (2)
Black or African American and Hispanic or Latina	0 (0)	1 (3)	1 (2)
White and Hispanic or Latina	3 (12)	1 (3)	4 (7)
White and Black or African American	2 (8)	1 (3)	3 (5)
**Income**			
USD 0–24,999	13 (50)	10 (34)	23 (42)
USD 25,000–49,999	12 (46)	16 (55)	28 (51)
USD 50,000+	1 (4)	3 (10)	4 (7)
**Education**			
HS diploma or less	8 (31)	4 (14)	12 (22)
Some college or associate degree	16 (62)	13 (45)	29 (53)
4-year college degree or more	2 (8)	12 (41)	14 (25)
**Participates in SNAP**	12 (46)	16 (55)	28 (51)
**Pregnant**	2 (8)	2 (7)	4 (7)
**Average number of children**	1.7	2.1	1.9

HS: high school; SNAP: Supplemental Nutrition Assistance Program; * One participant in the rural group selected “Prefer not to answer” for their race/ethnicity.

## Data Availability

The data presented in this study are available on request from the corresponding author. The data are not publicly available due to ethical and privacy issues of study participants.

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
