# Peer review of "“I Think That’s the Most Beneficial Change That WIC Has Made in a Really Long Time”: Perceptions and Awareness of an Increase in the WIC Cash Value Benefit"

_ijerph, 2022, doi:10.3390/ijerph19148671_

Round 1
Reviewer 1 Report
The article is clear, easy to follow and logically structured. The structure of the model is clear, easy to construct and well described. The argumentation and methodology in the paper is convincing.
Minor revision:
1. It is recommended to re-read the text to eliminate small typos. (e.g. line 332)
2. In the sbstract section, the full name of WIC should be given.
Reviewer 2 Report
Dear authors,
Thank you for your article and your research on this important topic.
This article is a report of a qualitative empirical study investigating WIC participants perceptions of the CVB increase during the COVID pandemic and barriers and facilitators to using the CVB. It uses online focus groups as data collection tool and thematic analysis for analysing the data. The study comes up with interestring results, as for example that CVB increase may improve fruit and vegetable intake. However, barriers to CVB and WIC benefit use may be limiting the potential impact. The study identified several barriers that might have political public health implications.
Line 20: For international readership I would suggest describing the meaning behind the WIC Cash Value Benefit (CVB) in the abstract as this might not be clear to every reader.
Introduction: The article opens with a broad introduction into the research field and embeds the study within the critical discourse of healthy nutrition of children. It points out to structural differences between rural and urban areas as well as to different opportunities for a healthy nutrition within various societal groups, as for example marginalized groups. It connects to prior research on WIC participants and the CVB increase and clearly identifies the research gap.
Methods: In the methods sections the authors clearly describe the procedure of participant recruitment, data collection and analysis.
Results: The results section provides some interesting new evidence of the difficulties of following nutrition advice with a limited budget and under health disparities. One of the most important results concerns the limited flexibility to pursue a healthy diet in accordance with the WIC recommendations on fruit and vegetable intake and variety. The authors reveal that before the pandemic the CVB amount was to less for the participants to be able to provide healthy meal options to their families and familiarize their children to healthy food options. The increase allowed for pursuing dietary patterns more closely aligned with their family’s preferences and WIC recommendations. The increase gave the participants freedom to include new fruits and vegetables into their diets and eat a more varied diet. This is a very interesting aspect that needs to be studied in the long term, and in comparison to changes in other food groups. The discussion includes valuable recommendations for future research and policy.
Reviewer 3 Report
The manuscript presents a WIC participant perception of increased WIC Cash Value Benefits (CVB) for fresh produce during the COVID-19 pandemic. The paper is interesting from qualitative point of view. I suggest the following issues to be addressed to improve the manuscript.
· 1. The authors should consider revising the title of the paper to better reflect the research/study. i.e., “Household perceptions and Awareness of an Increase in the WIC Cash Value Benefits for Fruits and Vegetables”
· 2. Author contributions are given at the end of the manuscript. Delete author names in the main body of the manuscript (line 129, 163, 175
· 3. In Table 1, the sample size in rural focus group and the number of responses in race/ethnicity group doesn’t match. Does it mean that one respondent didn’t disclose his/her race/ethnicity?
· 4. Authors have presented a qualitative analysis and mentioned in several places in the text “some/most participants”. Instead, authors should present a table/figure summarizing key variables and state the percentage/number of participants to enhance the clarity of their presentation.
